# Medical-Expert Eye Movement Augmented Vision Transformers for Glaucoma Diagnosis

Shubham Kaushal [1], Roshan Kenia[2], Saanvi Aima[2], Kaveri A. Thakoor[123]

[1]Data Science Institute, Columbia University, New York, NY, United States

[2]Department of Computer Science, Columbia University, New York, NY, United States

[3]Department of Ophthalmology, Columbia University Irving Medical Center, New York, NY, United States
{sk5118, rk3291, sa4166, k.thakoor}@columbia.edu

*Abstract*—intelligence (AI) in expediting glaucoma detection and enabling consensus. The Vision Transformer (ViT) model is a promising solution for this problem as it uses the self-attention mechanism to improve performance and interpretability. Furthermore, eye-tracking data provides valuable information about a clinician's decision-making process during the diagnosis of glaucoma using Optical Coherence Tomography (OCT) reports. In this study, two approaches were originated for incorporating eye-tracking data into the ViT's training process, using solely eye movement fixation order and attention-alignment loss. Fixation-order-informed (FOI) ViT models were found to perform better than the original ViT model, with fewer parameters and faster training. The use of attention-alignment in the ViT loss function resulted in improved performance when the effect of clinician-generated spatial attention was increased. The attention maps generated by these modified ViTs enabled interpretability and made the reasons for missed predictions more transparent especially for our FOI ViT model. Overall, these findings demonstrate the potential of using expert eye-tracking data to improve the performance of ViT models in glaucoma diagnosis.

*Index Terms*—attention, computer-aided diagnosis, eye-tracking, glaucoma, interpretability, optical coherence tomography, vision transformer

## I. Introduction

Glaucoma is a chronic eye disease that leads to optic nerve damage and vision loss if left untreated. It is the leading cause of irreversible blindness worldwide, affecting an estimated 80 million people in 2020, and its prevalence is expected to rise to 111.8 million by 2040 [1], [2]. The prevalence of glaucoma is higher among older adults, with an estimated 10% of individuals over 80 years old affected by the disease [3]. In the United States, it is estimated that more than 3 million people have glaucoma, and it is the leading cause of blindness among African Americans.

Optical coherence tomography (OCT) imaging is commonly used to diagnose glaucoma, but its interpretation can be challenging due to complexity of the images and variation in individual anatomy. Clinicians must carefully analyze OCT scans to identify structural changes that indicate glaucoma

Supported in part by an unrestricted grant to the Columbia University Department of Ophthalmology from Research to Prevent Blindness, Inc., New York, NY USA.

progression, which can be a subjective and time-consuming process. Moreover, there can be discrepancies in the interpretation of OCT scans among different clinicians [4]. As a result, there is a need for objective and reliable diagnostic tools to support clinicians in achieving accurate and efficient glaucoma diagnosis to expedite delivery of sight-saving treatment. Therefore, the development of automated methods for analyzing OCT scans can potentially improve the accuracy and consistency of glaucoma diagnosis.

Artificial Intelligence (AI) has emerged as a promising solution to improve glaucoma diagnosis. Machine learning algorithms can analyze large amounts of data and identify patterns that may be missed by human clinicians [5]. However, interpretability remains a crucial issue in the field. The 'black-box' nature of AI systems is a significant barrier that hinders their adoption in the clinic. Clinicians need to understand how AI models make decisions to trust their recommendations fully. Therefore, there is a need to develop AI models whose reasoning can be explained to clinicians [6]. This will enable clinicians to make informed decisions about patient care and improve the accuracy of glaucoma diagnosis.

Medical expert visual attention is a valuable tool for interpreting clinical decisions. In 1981, Carmody et al. [7] conducted one of the earliest eye-tracking studies in radiology, investigating the detection of lung nodules in chest X-ray films. The study revealed that individual radiologists' eye scanning strategies could impact the false negative errors in X-ray readings. Kundel et al. [8] collected eye-tracking data and found that 57% of cancer lesions were located within the first second of viewing. These findings suggest that eye movements of medical experts could help elucidate their decision-making process and enhance bioinspired AI. Experts' visual mechanisms that underlie their decision-making process, as reflected in their eye movements, can offer valuable insights for developing more interpretable and reliable AI partners for clinicians. By incorporating eye-tracking into the clinical workflow and analyzing the resulting data, AI models can be trained to better emulate the decision-making process of medical experts, leading to improved accuracy of diagnoses.

We propose two novel strategies that utilize Vision Trans-

former (ViT) models and eye-tracking data from clinicians for detecting glaucoma from OCT reports. The first strategy prioritizes computational efficiency by analyzing only the image patches that experts fixate on during glaucoma diagnosis from OCT images. The positional embeddings used in this approach are derived from clinician fixation order, rather than inherent image structure. The second strategy modifies ViT's attention mechanism by incorporating a loss function that aims to align its attention map with that of the clinicians. By employing these approaches, we seek to aid the diagnosis of glaucoma while also bolstering interpretability.

## II. BACKGROUND

### A. Eye Tracking and Deep Learning in Medical Imaging

In recent years, there has been a growing interest in exploring the potential of eye-tracking data in medical image analysis, particularly from the perspective of deep learning. One approach is to use eye-tracking data to aid in the annotation of medical images, which can be a time-consuming and labor-intensive task. For example, Stember and colleagues [11] used eye-tracking data and speech recognition to automatically label tumors in brain images. This approach was found to be faster and easier than the conventional manual annotation using mouse clicking and dragging.

Another area of research is investigating the relationship between human visual attention and deep learning models in medical image analysis. Mall and colleagues [12] explored this relationship in the context of mammography, studying the effectiveness of convolutional neural networks (CNNs) in finding cancer in mammography data. The authors also modeled the visual search behavior of radiologists looking for breast cancer using CNNs [13]. Li and colleagues [14] introduced an attention-based convolutional neural network (AG-CNN) for glaucoma detection, addressing the lack of attention mechanisms in existing CNN-based medical image recognition approaches. The authors established a large-scale attention-based glaucoma (LAG) database with 5,824 labeled fundus images and attention maps collected via simulated eye-tracking experiments done by ophthalmologists. The proposed AG-CNN includes an attention prediction subnet, a pathological area localization subnet, and a glaucoma classification subnet, visualizing features as localized pathological areas to enhance glaucoma detection performance.

Numerous studies have demonstrated the usefulness of eye-tracking data as a prior for deep learning models in medical imaging. Huang and colleagues [15] drew inspiration from the selective attention mechanism observed in human visual processing, which allows the cognitive system to learn from a limited number of training samples by selectively attending to task-relevant visual clues while ignoring distractors. In their work, the authors leveraged eye-gaze data in a limited data setting and developed a method that achieved superior performance in both 3D tumor segmentation and 2D chest X-ray classification tasks. The success of Huang and colleague's approach highlights the potential of eye-tracking data to guide deep learning models in medical imaging. By using eye-gaze data as a prior, deep learning models can focus on the most informative regions of the image, thereby improving efficiency and accuracy, especially in cases where training data is limited.

Jiang and colleagues [16] obtained eye-tracking data from clinicians performing diabetic retinopathy diagnosis and explored different ways to utilize the gaze maps. They evaluated the impact of gaze maps on the original fundus image through two image fusion methods and used a weighted gaze map to guide a neural network model's attention learning. Furthermore, they proposed an attention learning strategy that considers the difficulty and class of the image to improve model interpretability.

Similarly, Saab and colleagues [19] also introduced a method that integrates passively and inexpensively collected gaze data into a CAD system for medical image classification by transforming the gaze data into a rich source of supervision. They identified a set of gaze features and demonstrated that they contain class-discriminative information. They then proposed two methods for incorporating these gaze features into deep learning pipelines. Their findings revealed that their their method without task labels performed comparably to models trained with task labels. When task labels were available, their method exhibited improved performance over multiple baselines. This work demonstrates the potential of gaze data as a powerful tool for training deep learning models.

In fact, the study conducted by Ma and colleagues [20] stands as the sole endeavor to enhance the training of a ViT using eye-tracking data in medicine. Their model, known as eye-gaze-guided ViT (EG-ViT), involves masking out patches in the input radiology image that fall outside the radiologists' field of interest. Additionally, they introduce an extra residual connection in the last encoder layer. This not only resulted in improved performance but also enhanced the interpretability of the model. While such examples have focused on radiology, we are one of the first teams to introduce ophthalmologist viewing behavior on OCT images into the training of ViT models specifically for the detection of glaucoma. We incorporate eye-tracking data into the training process without modifying the ViT architecture itself. Therefore, our methodologies remain agnostic to ViT variants such as DeiT [21], PVT [22], TNT [23], Swin [24], and CSWin [25]. Our primary objective is to harness the potential of eye-tracking data for ViT training by exploring various approaches to integrate this data. We strive to develop techniques that strike a balance between interpretability, generalizability, accuracy, computational efficiency, and the preservation of the original model architecture.

## III. METHODS

Eye-tracking data was obtained from a group of 14 clinicians at the Harkness Eye Institute, Columbia University Medical Center, comprising of 9 residents and 5 glaucoma faculty/fellows. The clinicians were presented with 2 subsets of 20 OCT reports (10 glaucomatous and 10 healthy), one set fixed for all participants ('control') and the other randomly selected ('experimental') out of a total of 185 OCT reports,

with 121 reports being glaucomatous and 64 reports being healthy. During the analysis of the OCT reports, the clinicians wore a Pupil Labs Core (200-Hertz) eye-tracker device, which was mounted on their heads like glasses. They were asked to provide a score between 0 and 100 for glaucoma (0 indicating definite health and 100 indicating definite glaucoma) and to briefly summarize the OCT report features used in making their diagnoses. This study, Protocol AAAU4079, was approved by the Columbia University Irving Medical Center Institutional Review Board on 12/22/2022 and is in accordance with the tenets set forth by the Declaration of Helsinki. Informed consent was obtained from all study participants.

### A. Fixation-Order-Informed (FOI) ViT

In this section, we outline the steps involved in training a ViT model that incorporates the order of fixations made by clinicians on OCT reports during glaucoma diagnosis. Supp. Fig. 5 shows a flow-chart of both FOI and OGA ViT pipelines.

*1) Eye-tracking Data Preprocessing:* In our preliminary analysis, we observed that glaucoma faculty/fellows required on average approximately half as many fixations on OCT reports as residents to make correct glaucoma diagnoses [26]. Given that glaucoma faculty/fellows are able to make correct predictions faster (with fewer fixations) than residents, we chose to exclusively use eye-tracking data from glaucoma faculty/fellows (hereafter referred to as 'experts') in developing our FOI ViT model.

To mitigate errors and optimize accuracy/precision, we validated the eye tracker on a simple calibration task prior to data collection; we controlled room lighting, the display monitor and settings, and prevented head-mounting slippage by measuring and keeping nose-to-screen distance constant between participants. In spite of these efforts, the eye-tracker did at times produce noisy measurements due to minor variation in calibration, focus settings, and lighting conditions. In some instances, the number of fixations on a report was less than 10, and all fixations were concentrated in a relatively small region despite the clinician looking at different regions of the report. To eliminate these noisy trials, we retained eye-gaze patterns where the number of fixations was greater than 40 (equivalent to approximately 10 seconds) and the spatial variance was in the top 50%. We calculated spatial variance for any eye-gaze pattern by computing the covariance matrix between $X$ and $Y$ coordinates (which is a $2 \times 2$ matrix) and adding all entries, except one of the non-diagonal entries.

Since each of the five experts viewed one randomized experimental subset of 20 reports and one controlled subset of the same 20 reports each, we initially had 200 eye-gaze patterns (40x5). After limiting the eye-gaze patterns to those that corresponded to correct diagnoses (based on prior ground truth established by expert review of clinical and imaging data), we were left with 171 patterns. Finally, by applying the aforementioned spatial variance criteria, we were left with 31 eye-gaze patterns. These patterns were incorporated individually into different FOI ViT models, and the methodology for incorporating them into ViT training is described below.

*2) Image Preprocessing:* The original OCT reports were of size $2680 \times 5375$ pixels or larger; to decrease model processing time, we reduced each dimension of the input image to roughly one-fifth of its original value, making the dimension of each input image $500 \times 1000$ pixels irrespective of initial size. The pixel values of resized images were normalized using pre-determined mean and standard deviation values of $[0.485, 0.456, 0.406]$ and $[0.229, 0.224, 0.225]$, for R, G, and B channels, respectively. These values are consistent with those used for normalizing images while training the popular ResNet [27] CNN. Such image normalization ensures that the pixel values across images belong to a similar distribution, which leads to faster model convergence.

To facilitate input of eye-gaze data from clinicians, our fixation-order-informed ViT (FOI ViT) model requires images to undergo further pre-processing. Specifically, we select corresponding patches from the OCT report based on the clinician's eye-gaze pattern, and stack these patches in a row-wise raster fashion. The relative placement of patches is determined by the order in which the clinician looked at different parts of the report, with unviewed patches being dropped. The resulting processed image has dimensions of $h \times 1000$, where $h$ is the height of the image, equal to the number of fixations in the eye-gaze pattern, multiplied by the patch width of 20, divided by 1000, and then multiplied by the patch height of 20. As shown in Figure 1 (bottom, also enlarged in Supplementary Figure 6), the top-left patch represents the first patch viewed by the clinician, while the bottom-right patch (non-black) corresponds to the last patch viewed. Prior to being inputted into the FOI ViT model, this pre-processing step is performed on every image based on the corresponding eye-gaze pattern. Each of the 31 eye-gaze patterns are used in 31 different models, and each model applies its corresponding eye-gaze pattern to all 185 input images.

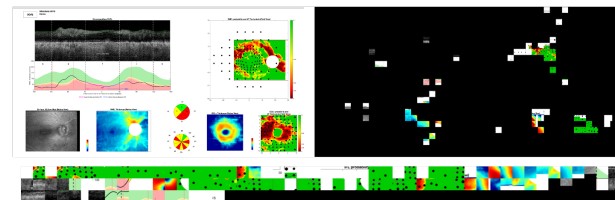

Fig. 1. Top-left: example original OCT report; Top-Right: example eye-gaze patches in context (with remaining OCT report blackened); Bottom: eye-gaze patches stacked in row-wise raster fashion, serving as input to FOI-ViT model. Enlarged version of this figure also in Supplementary Materials.

*3) Data Augmentation:* In the different experiments, the clinicians viewed OCT reports with varying appearances. During preprocessing, the images are resized to $500 \times 1000$ pixels to ensure consistent dimensions, and the colors are normalized as previously described. However, additional differences such as relative distances between sections, as well as the sizes and aspect ratios of those sections, must also be considered. Failure to account for these differences could result in the FOI ViT

model seeing a patch in some reports but not others, leading to insufficient information for making reliable predictions.

To solve the problem of differences in appearance between OCT reports, two types of data augmentations are used during image pre-processing. Firstly, the $X$ and $Y$ coordinates in the eye-gaze pattern are perturbed by 1 $unit$ probabilistically. Specifically, there is a 5% chance of incrementing and a 5% chance of decrementing any $X$-coordinate, and the same for any $Y$-coordinate in the gaze pattern; we call this data augmentation strategy 'jittering'. Secondly, the area of each patch's receptive field is increased probabilistically to ensure that an eye-gaze pattern is not restricted to a narrow region across different reports. With 80% probability, each patch has an area of 1 $unit^2$. With 15% probability, the patch area is 4 $unit^2$, and with 5% probability, it is 9 $unit^2$; we refer to this data augmentation strategy as varying 'receptive field'. Even though the receptive field of a patch may increase, it is down-sampled to be consistent with patch size of $20 \times 20$.

Supplementary Figure 1 shows patches extracted using eye-gaze data in their original positions on the OCT report (top) and with data augmentations applied (bottom). The first difference one can notice is that positions of many patches are different between top and bottom. Secondly, one can observe that neighboring patches in Supp. Fig. 1 (top) are continuous and join seamlessly. On the other hand, neighboring patches in Supp. Fig. 1 (bottom) do not join seamlessly, and this is because the patches may have different receptive fields (1, 4, or 9 $unit^2$). Overall, this dynamic augmentation allows the FOI ViT model to 'see' regions that are near but along the gaze-pattern, accounting for small differences in relative distances, sizes, and aspect ratios between reports.

*4) Model Training:* To evaluate the impact of integrating fixation order into ViT training, we employed a control ViT model and several FOI ViT models, each utilizing a different eye-gaze pattern. The primary difference between control and FOI ViT models was in the pre-processing of their input images and the setting of their positional embeddings. Despite these differences, both models shared certain similarities. For instance, we chose a patch size of $20 \times 20$ for both models, as this size is the smallest common divisor of the resized image dimensions ($500 \times 1000$) that is greater than 16, which is the patch size used in the original ViT paper [28]. It is worth noting that we did not select a smaller patch size, as this would have resulted in a higher number of patches and increased computational complexity. Additionally, we used the hyperparameters shown in Table I for both models.

The differences in input image pre-processing and positional embedding setting between the control and FOI ViT models were motivated by the aim to investigate the impact of gaze-data on FOI ViT performance. In natural language processing (NLP), transformers use sinusoidal positional embeddings to encode positional information of tokens in a sequence [29]. Similarly, in ViT, positional embeddings encode information about the position of a patch in the image. Positional embeddings can be visualized as vectors that shift close patches in the image to close positions in the embedding

TABLE I
HYPERPARAMETER CHOICES FOR CONTROL AND FOI ViT MODELS

| | |
|---|---|
| Patch Size | $20 \times 20$ |
| Embedding Dimension | 1024 |
| Depth | 3 |
| Number of Heads | 8 |
| MLP Dimension | 1024 |
| Dropout Rate | 0.1 |
| Embedding Dropout Rate | 0.1 |
| Optimizer | Adam |
| Weight Decay | $10^{-4}$ |
| Learning Rate | $10^{-4}$ |
| Number of Epochs | 15 |
| Batch Size | 10 |
| Expert Participants | 5 |
| Gaze-Image Samples Per Model (for testing accuracy) | 185 |
| Number of Unique FOI-ViT Models | 31 |

space. The authors of the original ViT paper [28] made their positional embeddings learnable, instead of sinusoidal, which are non-learnable. They found that the model was able to learn appropriate positional embeddings based on the inherent image structure. The control ViT model's learnable positional embeddings allow it to learn relative positions between patches in the report. This information can be leveraged to identify which parts of the report are crucial to making a glaucomatous prediction.

On the other hand, our FOI ViT model uses non-learnable sinusoidal positional embeddings, akin to those in NLP transformers. In NLP, the language's grammatical structure determines the order in which tokens appear in an input. Similarly, for our FOI ViT models, the expert eye-gaze pattern determines not only the patches that the model can see but also the order in which patches appear in the input image. Consequently, non-learnable positional embeddings constrain/refine the FOI ViT model to "see" the report in the same manner that clinicians did.

### B. Attention-Aligned Loss in Vision Transformers

Our second strategy for augmenting the ViT with expert eye movement data involved guiding the transformer to "learn" to match certain aspects of clinician eye movement behavior. The ViT's self-attention mechanism gives insight into how the model mimics the human visual system's approach for deciding what to learn from the most while training [31]. We attempted to guide the self-attention of the ViT to high-density fixation regions of clinicians, thereby learning human expert priors to improve the ViT's accuracy and interpretability.

To achieve this, we introduced an attention-alignment term in the loss function of the ViT model. In addition to a regular classification cross-entropy loss term, $L_{classification}$, our expert attention-aligned ViT incorporates an additional "attention-alignment" (AA) term into the loss function (as shown in Equation 1), which measures the similarity between the ViT attention weights and clinician eye-tracking patterns. This cross-entropy distance between medical-expert attention and ViT self-attention ($L_{AA}$) is weighted by an additional $\alpha$

hyperparameter, which determines the importance of alignment in model training. $\alpha$ is determined manually. When $\alpha$ is set to 0, it indicates no clinician knowledge involved; when $\alpha$ is set to 1, it indicates only clinician eye fixation attention (no ViT self-attention) is involved. By employing this approach, we seamlessly integrate rich human visual attention information into the loss function of the ViT. Leveraging expert eye movement data as a guide, we instruct the ViT to align its attention weights with the fixation points identified by clinicians during their diagnostic process. This bears a resemblance to knowledge distillation [32], yet our exclusive teacher is composed solely of clinician gaze information.

$$Loss = (1 - \alpha) \times L_{classification} + \alpha \times L_{AA} \quad (1)$$

*1) Eye-tracking and OCT Image Preprocessing:* To quantify alignment between ViT attention and expert clinician attention, we converted the clinician eye-tracking data and ViT self-attention matrix into matrices of the same dimension, in which each grid corresponds to a patch in the original OCT scan. The ViT attention matrix is computed by taking the self-attention outputs from the ViT and marginalizing the attention directed into each patch. The eye-tracking attention matrix (hereafter referred to as 'heatmap') is computed as described in the next paragraph.

Within our dataset, a single OCT report may be reviewed by multiple clinicians, resulting in various sets of eye-tracking data for the same report. In order to avoid creating multiple heatmaps for a single report, we aggregate all the gaze data of clinicians who viewed a given report via a method we call 'patch bucketing', creating a global heatmap for that OCT report derived from all the clinicians who viewed it. For every clinician who assessed the same report (on average a given report was viewed by about 3 clinicians), we standardize the X and Y positions (ranging from 0.0 to 1.0) of their fixation locations and group them together. We then create a 14x14 heatmap by multiplying each normalized X and Y value by 14; to aggregate all clinician fixation data, we increment the quantized patches within the 14x14 heatmap that correspond to each clinician's fixations in a given OCT report. The resulting heatmap accumulates counts where clinicians fixated most on that OCT report. Finally, we normalize this global heatmap to produce a weighting between 0.0 to 1.0 for each region. This technique results in a consistent, 'global' heatmap representation capturing all clinician eye movement data for a given OCT report while training the ViT.

*2) Model Information:* The model, a pretrained transformer using Google's *vit-base-patch16-224-in21k* transformer, was trained (fine-tuned) on 185 OCT scans and their corresponding eye-tracking data.

## IV. RESULTS

### A. Fixation Order Informed ViT

To ensure a fair comparison of performance, we conducted 5-fold cross-validation five times (yielding 25 model evaluations) on both the control ViT model and the 31 FOI ViT models, using identical hyperparameters. The optimal model for each fold was determined using validation loss as the measure of performance. The results from training these models are summarized in Table II, where each column represents a metric averaged over the 25 optimal models from each fold. In order to provide a clear picture of the relative performance of the FOI ViT models, we have presented the results for two models, FOI ViT-4 and FOI ViT-20, which represent the best and worst performing FOI ViT models, respectively. We also provide results in Supplementary Table I of models generated by ablating either our jittering or receptive field data augmentation approaches (or both) to show the relative contribution of these strategies on model performance.

To assess whether differences in performance were statistically significant, we conducted Mann-Whitney U tests between all models. Our choice of performance measure was the F1-score, as it is robust to class imbalance, which is a common occurrence during cross-validation due to our dataset containing 121 glaucomatous reports and only 64 healthy reports. Our analysis revealed that out of the 31 FOI ViT models, four performed significantly worse than the control ViT model, with FOI ViT-20 being the poorest performer (refer to Supp. Table 1). Conversely, the remaining 27 FOI ViT models performed comparably or slightly better than the control ViT model, with FOI ViT-4 emerging as the best performer among them. Supp. Table 1 ablation tests show that, while presence of both jitter and receptive field augmentation strategies together did help to improve FOI ViT-4's performance, presence of only one augmentation strategy had negligible impact on performance.

### B. Attention-Aligned Loss in Vision Transformers

Figure 2 demonstrates the alignment score (shown in Equation 1) measured over about 30 epochs of training as well as corresponding expert-aligned ViT attention outputs. The alignment score here represents the cross entropy between attention outputs and clinician fixation data. A decreasing loss score indicates that the transformer is aligning its attention to be more similar to clinician gaze patterns. Table II shows glaucoma classification performance of attention-aligned (hereafter referred to as 'OGA', Ophthalmologist-Gaze-Augmented) ViT models at varying values of $\alpha$.

## V. DISCUSSION

Table II shows collective performance (mean F-1 score, precision, and recall) of our best FOI ViT models, best OGA ViT models (with optimal $\alpha$ values), and multiple baseline models using CNN and ViT architectures with and without gaze overlaid on OCT reports. Comparison with these baselines serves to showcase the glaucoma detection accuracy improvement afforded by incorporation of clinician eye tracking data into ViT training.

### A. Fixation-Order-Informed ViT

*1) Efficiency and Accuracy:* Our findings demonstrate that the incorporation of eye-tracking data from clinicians enables

MEAN F-1 SCORE, MEAN PRECISION, AND MEAN RECALL GLAUCOMA DETECTION PERFORMANCE FOR FOI VIT AND ATTENTION-ALIGNED (OGA) VIT VS. BASELINE CNN AND VIT ARCHITECTURES WITH AND WITHOUT GAZE DATA OVERLAID ON OCT REPORT INPUTS

| Model and Input | # of Trainable Params | Image Resolution | Mean F1-Score | Mean Precision | Mean Recall |
|---|---|---|---|---|---|
| **FOI ViT-4** | **13M** | **hx1000, h $\ll$ 500** | **0.918** | **0.924** | 0.824 |
| OCT + Gaze + ViT | 16.4M | 500x1000 | **0.922** | **0.928** | **0.915** |
| OCT + Gaze + ResNet50 | 23.5M | 500x1000 | **0.908** | **0.921** | **0.901** |
| Control ViT (No Gaze) | 16M | 500x1000 | 0.898 | 0.917 | 0.883 |
| FOI ViT-20 | 16M | 500x1000 | 0.836 | 0.837 | 0.849 |
| **OGA-ViT (alpha=0.05)** | 85.8M | **224x224** | **0.920** | **0.960** | 0.896 |
| OGA-ViT (alpha=0.01) | 85.8M | 224x224 | 0.897 | 0.898 | 0.905 |
| OGA-ViT (alpha=0.1) | 85.8M | 224x224 | 0.888 | 0.900 | 0.899 |
| OGA-ViT (alpha=0), No Gaze | 85.8M | 224x224 | 0.866 | 0.875 | 0.882 |
| OCT + ResNet50 (No Gaze) | 23.5M | 224x224 | 0.860 | 0.888 | 0.840 |

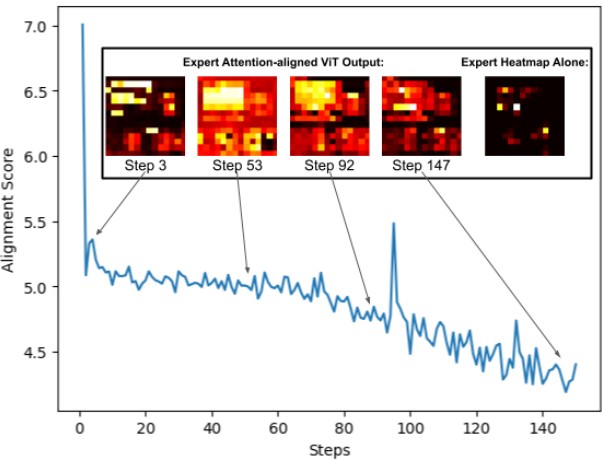

Fig. 2. Alignment loss between ViT self-attention and clinician global heatmaps for $\alpha = 0.05$. A lower score indicates that the ViT and clinician are focusing on the same regions. There are approximately 5-6 steps in each epoch. The inset shows heatmaps used to calculate attention alignment. Inset from left to right: attention-aligned (OGA) ViT outputs over time with $\alpha = 0.05$ and $t = 3, 53, 92$ and 147 steps, respectively. Note that the attention output at $t = 147$ steps highlights circumpapillary b-scan, RNFL, and GCL regions, consistent with the 'CU-Method' used by clinicians to diagnose glaucoma at CUIMC. Inset at far right: attention of cumulative clinician fixation data.

the development of FOI ViT models that perform with slightly higher accuracy (average F1-score of 0.918 for FOI ViT-4) compared to ViT models that do not use such data (average F1-score of 0.898 for control ViT). This outcome is particularly noteworthy given that almost all FOI ViT models observe less than 10% of each report, as each model is limited to only seeing the regions where a clinician's fixations landed; FOI ViT-4 in particular has an image resolution of h x 1000, where h is much smaller than 500. Moreover, FOI ViT models have around 13 million model weights (trainable parameters), while the control ViT model has approximately 16 million model weights, because the latter contains learnable positional embeddings. Due to the smaller input image size and fewer parameters, the FOI ViT model's training speed is 50% faster than that of the control ViT model.

Our FOI-ViT model also achieves glaucoma detection accuracy with comparable accuracy and higher efficiency (fewer model parameters) than baseline ViT (OCT + Gaze + ViT) and CNN-based (OCT + Gaze + ResNet50) architectures taking as input full OCT reports overlaid with (or without) clinician gaze as a heatmap (example gaze-overlaid image shown in Supplementary Figure 4); results of these baseline models are also shown in Table II. These results underscore the significance of eye-tracking data. Despite observing a significantly smaller portion of the OCT report and having fewer parameters, the FOI ViT model performs comparably to the larger and slower-to-train control ViT model as well as the baseline ViT and CNN architectures. Our findings indicate that the utilization of eye-tracking data, especially when using the fixation order in eye-gaze, can function as an efficient prior. Our introduction of jitter and receptive field data augmentation strategies also played a role in improving FOI ViT performance, as demonstrated in Supp. Table 1.

The poor performance of FOI ViT-20 in spite of utilizing gaze may arise from the fact that the gaze pattern for this model came from a trainee (with 3 or less years of experience), whereas the gaze pattern for our best FOI ViT-4 came from a glaucoma faculty member with 30 or more years of experience. This suggests that the variance in expertise levels among the participants could have played a role in the observed difference in model performance.

Statistically, we observed an effect size (Cohen's d) of 0.499 (based on FOI ViT-4 vs. control ViT F1 scores and standard deviations from Table II), with F-1 score increasing from 0.898 (control ViT) to 0.918 (FOI-ViT) on average (at 80% power, $\alpha$ = 0.05), indicating a sample size of 130 would be sufficient (65 in each class). After pre-processing, we analyzed 31 unique eye-gaze patterns applied to 185 OCT reports, yielding 185 gaze-image samples for each model assessed, providing a well-powered sample.

*2) Interpretability:* The attention maps generated by ViT models provide valuable insights into the areas of the input that the model attends to when making predictions. For example, Supplementary Figures 2 and 3 display the attention maps for the best and worst performing FOI ViT models, respectively. By analyzing these attention maps, researchers can gain a deeper understanding of why some eye-gaze patterns perform better than others. For example, Supp. Fig. 3 shows that the

FOI ViT-20 model's suboptimal performance is because it assigns higher importance to only one patch (yellow patch in the top-left is on a vitreous region of circumpapillary b-scan with no tissue), whereas FOI ViT-4 (Supp. Fig. 2) assigns high attention values to several patches corresponding to retinal nerve fiber layer (RNFL) and ganglion cell layer (GCL) regions, consistent with the systematic viewing 'algorithm' (known as the 'CU-Method') taught to ophthalmologists at CUIMC.

In addition to shedding light on the model's decision-making process, attention maps generated by high-performing FOI ViT models, such as the one displayed in Supp. Fig. 2, can be used to discern the relative importance of various OCT-report regions and compare them to regions that clinicians observe to learn potentially new OCT biomarkers picked up by the AI. Furthermore, when ViT models make incorrect predictions, researchers can use these attention maps to diagnose the reasons for these mistakes both for the AI or a medical trainee. For example, researchers can determine if the model assigns higher importance to an incorrect region or if the corresponding clinician eye-gaze pattern does not cover relevant regions.

### B. Attention-Aligned Loss in Vision Transformer

Our results with the attention-aligned loss function indicate that there are effects on accuracy with different levels of $\alpha$. As demonstrated in Figure 2, model accuracy improves when $\alpha$ is set to 0.05 compared to $\alpha$ of 0, 0.01, or 0.1, indicating a moderate expert attention contribution optimizes our attention-aligned ViT model.

Observing Figure 2, we see evidence of our model's medical-expert alignment process in action. The downward trend of the graph indicates that attention alignment is successfully altering the ViT model's behavior. We can also visualize the two inputs used to calculate the alignment score. On the far right of the inset, we see the aggregated clinician fixation data converted into a heatmap, where brighter patches represent OCT-report areas which have been fixated more often. On the left side of the Figure 2 inset, we see the ViT's attention-aligned outputs as they change over time. During training, the model observed samples of global clinician heatmaps and used these to appropriately adjust its own self-attention outputs. In this instance, the clinicians focused mainly on the top left and bottom right of the OCT scans. Whereas, the attention-aligned ViT focused attention towards the top left, top right, and bottom right of the report as training time increased, representing the b-scan, RNFL, and RGCL regions of the OCT report, consistent with the 'CU-method' used by clinicians at CUIMC to diagnose glaucoma. Furthermore, our 'patch bucketing'/global heatmap generation approach, while reducing sample size for training, enabled the aggregation of fixation information from multiple experts on one heatmap, enhancing our OGA ViT models' training consistency by ensuring aligned expert attention guidance for a given OCT input.

Thus, the effect of attention-alignment here serves an important role. By blending regions from the clinician fixations with areas that the vision transformer pinpoints as important, we utilize existing medical domain expertise while concurrently discovering new AI-learned regions. Such interplay bridges the clinician's existing prior knowledge with novel ViT-based insights, generating suggestions for clinicians while improving performance for deep learning models.

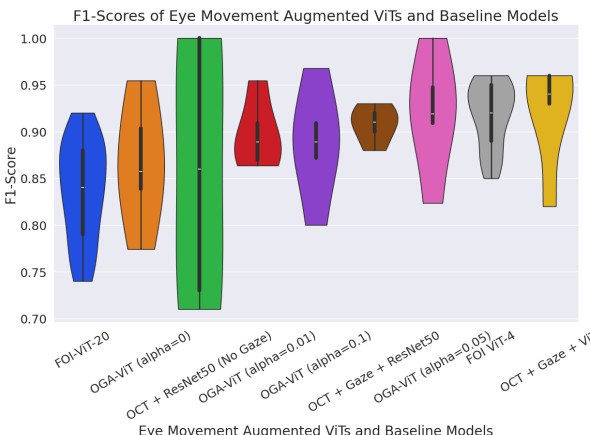

Fig. 3. Violin Plot Comparing FOI, OGA, and Baseline Models.

### C. Comparing Models with and without Gaze Augmentation

The violin plot in Figure 3 demonstrates the advantageous impact of integrating gaze data into model architectures, exemplified by notable improvements in median performance going from left to right in the figure. Models incorporating gaze data, such as FOI ViT-4, CNN, and ViT with overlaid gaze, and OGA ViT (with $\alpha$ of 0.05), consistently outperform their counterparts that lack eye-tracking information. This lower performance is particularly evident for the ResNet50 (without gaze), FOI ViT-20, and the attention-unaligned ViT model (with $\alpha$ of 0).

The width of the violin plots also serves as a visual gauge for the variability in outcomes. Models relying on eye-tracking data exhibit a greater frequency of data distributed closer to the median result values, indicative of heightened consistency. Moreover, these models yield higher F1-scores, showcasing their reliability and efficacy. In contrast, models that do not use gaze data not only demonstrate lower performance but also showcase inconsistent results, exemplified by their narrower frequency about the median. The incorporation of eye-tracking data emerges as a factor in enhancing both the robustness and overall accuracy of these models.

### VI. CONCLUSION

In conclusion, our study demonstrates the potential of using eye-tracking data to augment the training process of AI models for diagnosing glaucoma. The use of fixation-order-informed (FOI) ViT models and attention-alignment loss in ViT models have shown promising results, with better performance than the original ViT model while using fewer parameters/smaller

input image resolution and training faster as well as offering more interpretability/mechanistic transparency. Our findings suggest that increasing the effect of clinician-generated spatial attention in ViT models (OGA-ViT) can improve performance, highlighting the importance of using eye-tracking data in the ViT training process.

Moreover, our research presents a novel approach to incorporating eye-tracking data as a prior for training AI models, which can be extended to other types of medical reports and diseases. The ability to develop interpretable models that can provide new insights while also learning from experts to gain domain knowledge is crucial for the future of AI in healthcare. By doing so, we can improve accountability for both AI and clinicians and promote a better understanding of how AI systems make decisions. Just as this work builds on past work using ophthalmology resident gaze data alone [33], as we move forward, we anticipate that extending these models to larger datasets, with more reports and eye-tracking data, will further improve their effectiveness.

In future work, we aim to generalize our attention-aligned loss approach by capturing both fixation duration and sequence order of multiple experts by fully leveraging ViT's self-attention matrix and multiple heads. With the continued development of AI in healthcare, there is great potential for the use of eye-tracking data to augment the training process of AI models and improve clinical decision-making by enhancing the accuracy, efficiency, and interpretability of diagnoses made by 'medical-expert–AI teams', leading to better patient outcomes.

## ACKNOWLEDGMENTS

The authors would like to thank Jeffrey M. Liebmann and George A. Cioffi for their oversight and guidance, Ari Leshno for data sharing, Sophie Gu for help in recruiting participants for this study, and Royce W.S. Chen for valuable discussions.

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
