# Supplementary Materials: Medical-Expert Eye Movement Augmented Vision Transformers for Glaucoma Diagnosis

Shubham Kaushal [1], Roshan Kenia[2], Saanvi Aima[2], Kaveri A. Thakoor[123]
[1]Data Science Institute, Columbia University, New York, NY, United States
[2]Department of Computer Science, Columbia University, New York, NY, United States
[3]Department of Ophthalmology, Columbia University Irving Medical Center, New York, NY, United States
{sk5118, rk3291, sa4166, k.thakoor}@columbia.edu

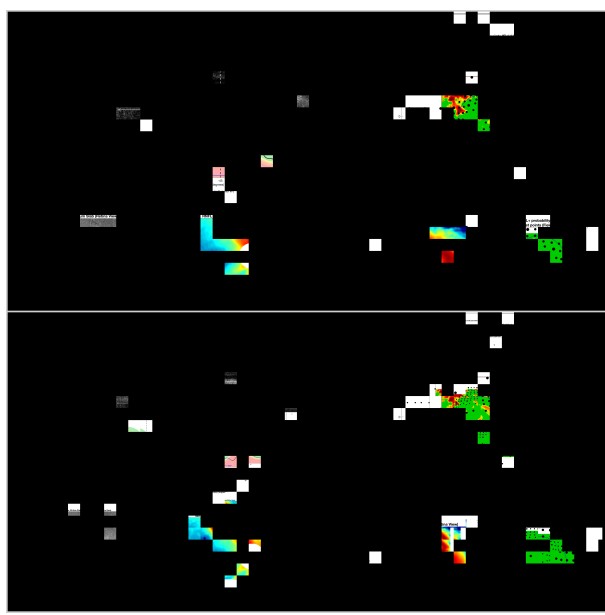

Fig. 1: Patches from eye-gaze before (top) and after (bottom) applying jitter and receptive field data augmentations.

TABLE I: Control ViT, Best (FOI ViT-4) and Worst (FOI ViT-20) FOI ViTs, and Ablation Tests of Jitter and Receptive Field Data Augmentation Strategies on FOI ViT-4

| Model | Val Loss±Std Dev | Val F1±Std Dev | Val Acc±Std Dev |
|---|---|---|---|
| Control ViT | 0.265±0.0925 | 0.898±0.0434 | 0.877±0.0512 |
| **FOI ViT-4 + jitter + RF** | **0.260±0.0663** | **0.918±0.0347** | **0.893±0.0438** |
| FOI ViT-4 - jitter + RF | 0.247±0.0855 | 0.894±0.0502 | 0.876±0.0476 |
| FOI-ViT-4 + jitter - RF | 0.250±0.0637 | 0.889±0.0484 | 0.868±0.0467 |
| FOI-ViT-4 - jitter - RF | 0.247±0.0654 | 0.894±0.0343 | 0.876±0.0347 |
| FOI ViT-20 + jitter + RF | 0.374±0.0685 | 0.836±0.0527 | 0.798±0.0625 |

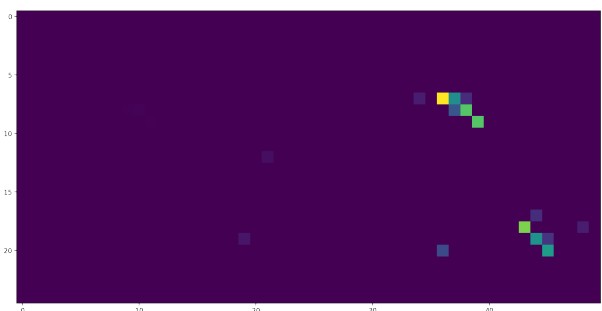

Fig. 2: Attention map from FOI ViT-4.

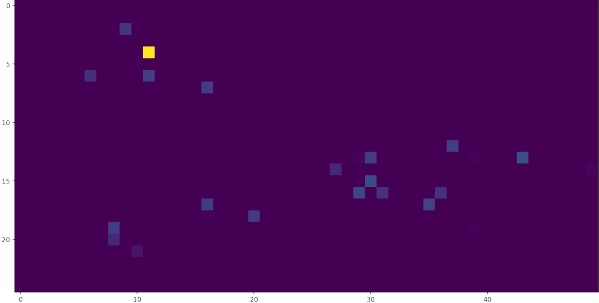

Fig. 3: Attention map from FOI ViT-20.

Supported in part by an unrestricted grant to the Columbia University Department of Ophthalmology from Research to Prevent Blindness, Inc., New York, NY USA.

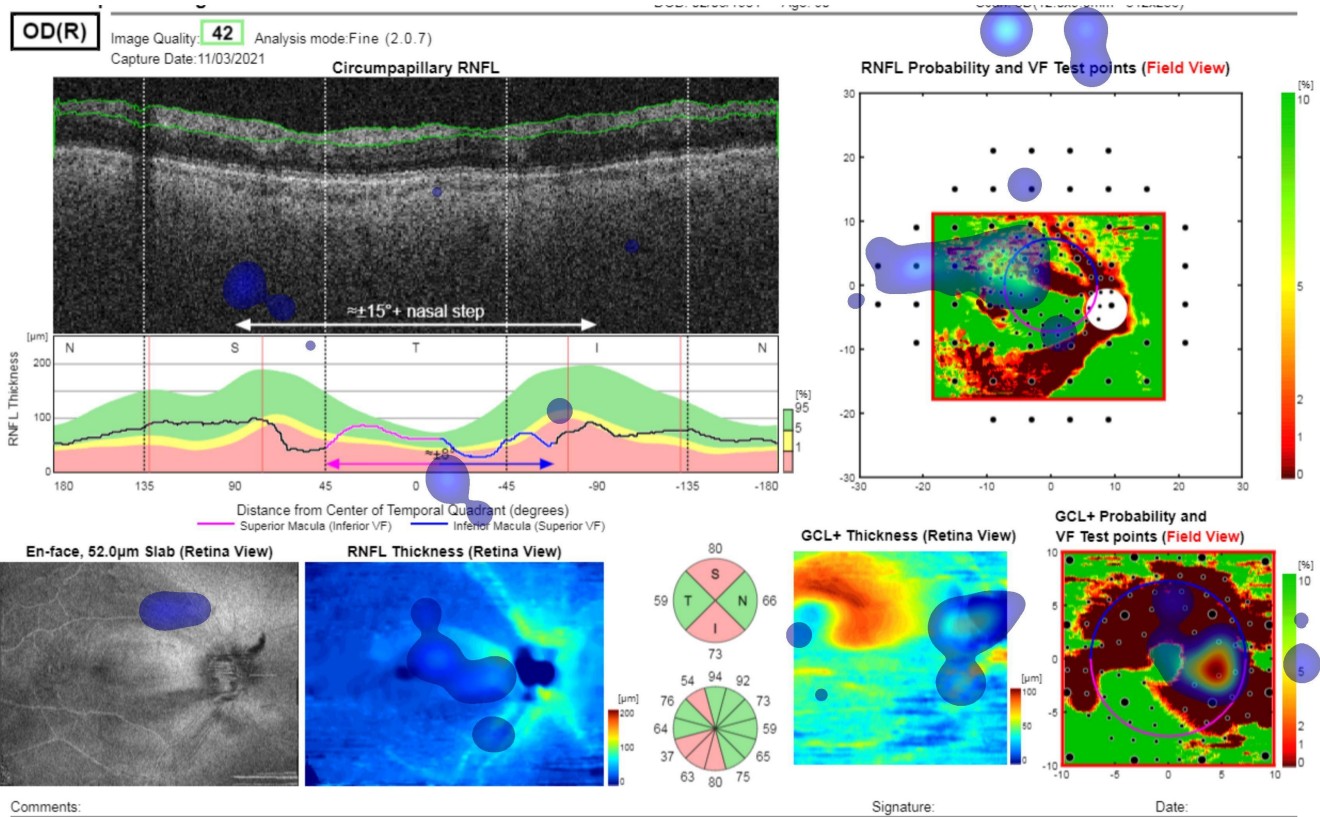

Fig. 4: To generate the gaze overlaid images, we took the normalized x and y values of each expert fixation location and multiplied them by the respective width and height values of the OCT report. We then plotted each fixation by convolving it with a Gaussian kernel (mean of 200ms, standard deviation of 6ms). We overlaid the same eye-tracking sequence used in our best performing FOI ViT-4 model on all images for ResNet50 with gaze and ViT with gaze. Areas fixated longer/more often are depicted with warmer colors (red/yellow), while regions fixated less often/for shorter duration are depicted by colder colors (blue/violet).

---

**Algorithm 1** Fixation-Order-Informed Vision Transformer (FOI ViT)

---

1: **function** FOIVIT($OCT\_report, eye\_gaze\_pattern$)
2:     $patches \leftarrow$ SELECTRELEVANTPATCHES($OCT\_report, eye\_gaze\_pattern$)
3:     **for** each $patch$ in $patches$ **do**
4:         $patch \leftarrow$ JITTERING($patch, prob = 0.05$)
5:         $patch \leftarrow$ VARYRECEPTIVEFIELD($patch$)
6:         **if** RANDOM($0, 1$) $\leq 0.80$ **then**
7:         $patch.receptive\_field \leftarrow 1$ unit²
8:         **elseif** RANDOM($0, 1$) $\leq 0.95$ **then**
9:         $patch.receptive\_field \leftarrow 4$ unit²
10:         **else**
11:         $patch.receptive\_field \leftarrow 9$ unit²
12:         **end if**
13:     **end for**
14:     $stacked\_patches \leftarrow$ STACKPATCHES($patches, eye\_gaze\_order$)
15:     $focus\_areas \leftarrow$ SELECTFOCUSAREAS($stacked\_patches, eye\_gaze\_pattern$)
16:     $control\_ViT \leftarrow$ CREATECONTROLVIT($focus\_areas, learnable\_positional\_embeddings$)
17:     $FOI\_ViT \leftarrow$ CREATEFOIVIT($focus\_areas, sinusoidal\_positional\_embeddings$)
18:     **return** $FOI\_ViT, control\_ViT$
19: **end function**

---

**Algorithm 2** Attention-Aligned (OGA ViT)

1: **function** ALIGNMENTLOSS($OGA\_ViT, OCT\_report, label, clinicians\_eye\_gaze\_pattern$)
2:     $aggregate\_eyemovt\_heatmap \leftarrow$ INITIALIZE($height = 14, width = 14$)
3:     **for** each $eye\_gaze\_pattern$ in $clinicians\_eye\_gaze\_pattern$ **do**
4:         **for** each $gaze\_coordinate$ in $eye\_gaze\_pattern$ **do**
5:             $gaze\_coordinate \leftarrow$ MODULO($gaze\_coordinate, n = 14$)
6:             $aggregate\_eyemovt\_heatmap \leftarrow$ INCREMENT($aggregate\_eyemovt\_heatmap, gaze\_coordinate$)
7:         **end for**
8:     **end for**
9:     $aggregate\_eyemovt\_heatmap \leftarrow$ NORMALIZE($aggregate\_eyemovt\_heatmap, low = 0, high = 1$)
10:    $prediction, attention \leftarrow$ OGA_ViT($OCT\_report$)
11:    $attention \leftarrow$ SUM($attention, dim = 1$)
12:    $detection\_loss \leftarrow$ CROSS_ENTROPY($prediction, label$)
13:    $alignment\_loss \leftarrow$ CROSS_ENTROPY($aggregate\_eyemovt\_heatmap, attention$)
14:    $loss \leftarrow$ WEIGHTED_SUM($detection\_loss, 1 - \alpha = 0.95, alignment\_loss, \alpha = 0.05$)
15:    **return** $loss$
16: **end function**

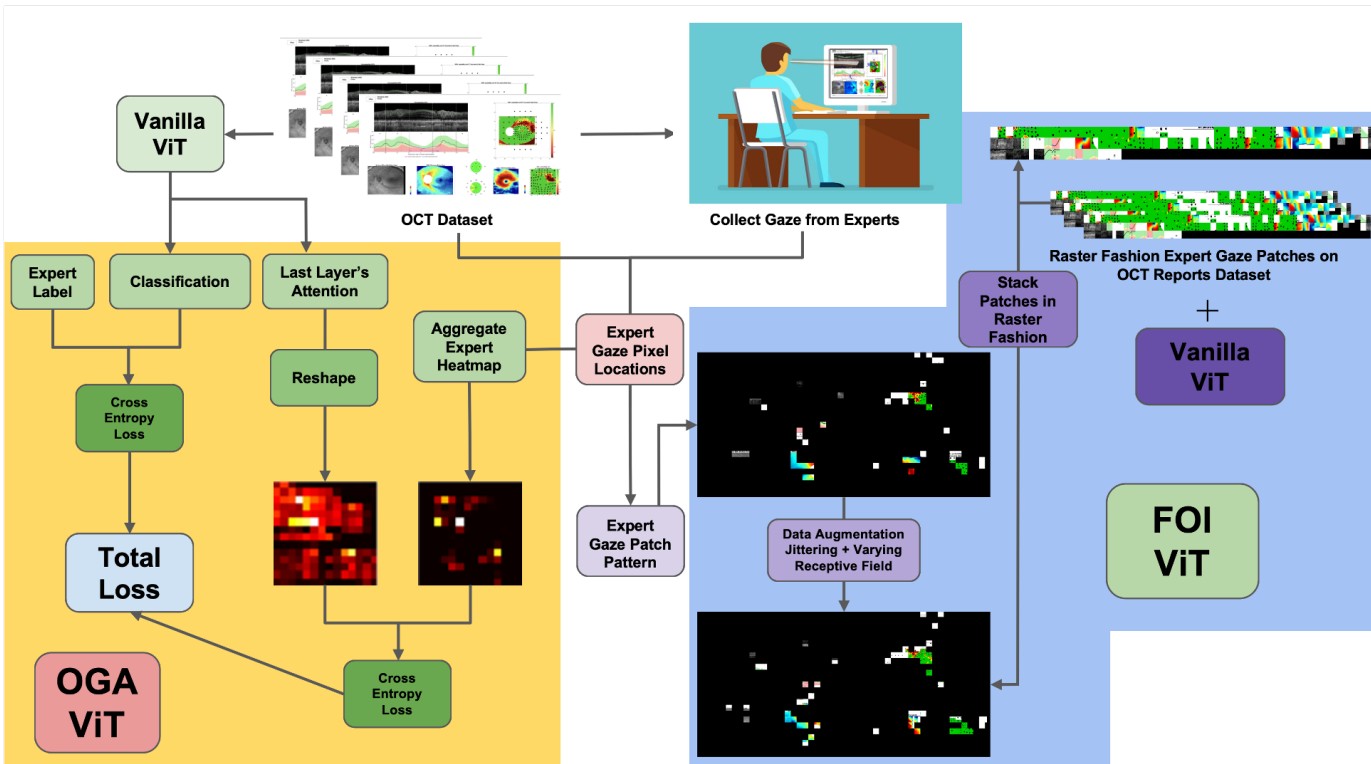

Fig. 5: Flow chart showing distinctions between FOI and OGA ViT algorithms presented in main paper. FOI ViT does not modify the main ViT architecture and only modifies the input image, which is composed only of the raster-stacked patches fixated by the expert ophthalmologist in the order in which the patches were fixated (blue background, right side of figure). In contrast, OGA ViT modifies the ViT loss function to include a term representing the alignment between the ViT's self-attention and aggregated expert eye-movement attention (yellow background, left side of figure).

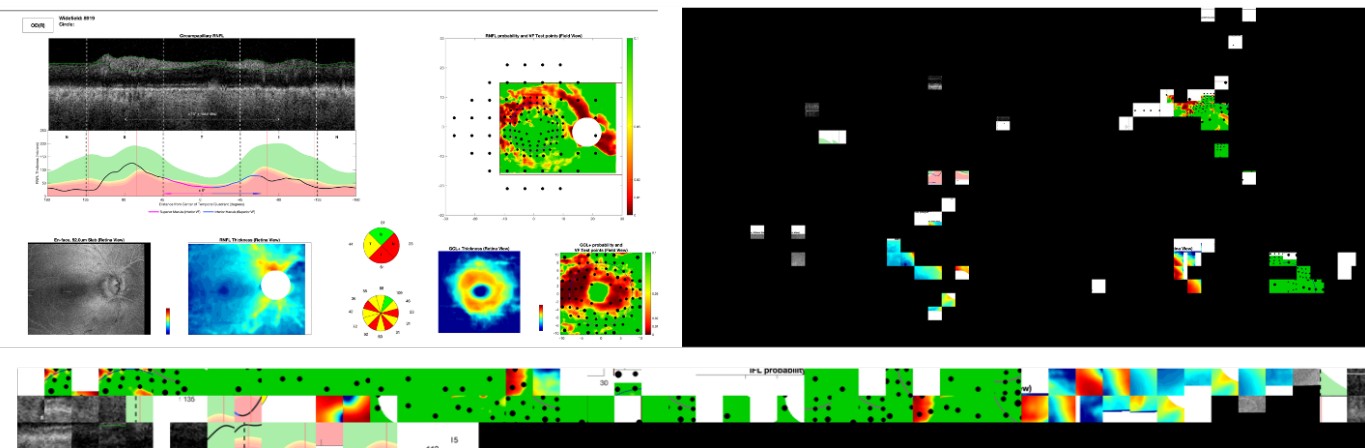

Fig. 6: Top-left: example original OCT report; Top-Right: example eye-gaze patches in context (with remaining OCT report blackened); Bottom: eye-gaze patches stacked in row-wise raster fashion, serving as input to FOI-ViT model.