# OpenReview forum: "Medical-Expert Eye Movement Augmented Vision Transformers for Glaucoma Diagnosis"
_IEEE.org/EMBS/BHI/2024/Conference — IEEE BHI'24_

### Official Review · Reviewer_QobH · 2024-08-06
**Interesting, well described work, further validation would be beneficial**

**Overall Rating:** 7
**Confidence:** 1

**Other Quality Metrics:**

(a) Clarity of writing: GREAT
(b) Clinical Significance: GOOD
(c) Methodological Novelty: GOOD
(d) Experiments and Results: FAIR

**Questions For The Authors:**

1) Please, better describe your dataset: the numerosities are not very clear to me, especially after the selection of only specialized practioners.
2) Please, reorganize Fig.1 and increase its size: at present it is impossible to read
3) I appreciate the idea of a graphical description of the pipeline (I think it's very beneficial), so I suggest concentrating on the improvement and increase of schematicity of Fig 1
4) Please, besides the meaning of alpha = 0, add the maximum value this parameter can reach, to orient the uninformed reader about your selected value
5) I would increase the number of folds, and add further metrics of evaluation: in diagnosis-aided tools it's crucial to distinguish between precision and recall

**Strengths:**

- thorough analysis of the background and well-motivated research
- very well described and discussed

**Summary Of The Paper:**

the authors propose the integration of experts gaze dynamics in a vision transformer based model to detect glaucoma

**Weaknesses:**

- limited dataset, explanation of the components is not very clear
- Figures are difficult to read, disproportioned (Fig.1 is too small, text of Fig.3 is too small), lacking information (Fig.3)
- I don't think such a thourough description of cross-validation is needed, as it is a rather basic concept. I would instead better describe the dataset, maybe using figures, using that space
- The results lack the precision and recall metrics, which are crucial in this kind of application

---

### Official Review · Reviewer_WUAf · 2024-08-16
**Review comment**

**Overall Rating:** 7
**Confidence:** 4

**Other Quality Metrics:**

(a) Clarity of writing: great
(b) Clinical Significance; great
(c) Methodological Novelty; great
(d) Experiments and Results, excellent

**Questions For The Authors:**

1. As presented in Sec III. All data used to train the mode is from only 5 experts. The data source is insufficient to support the generalizability of the proposed method.

2. In Sec.V.  The key idea of this paper is to make use of expert knowledge as prior knowledge for better model performance. It's better to compare with a learnable positional embedding model to prove the effectiveness of your idea.

3. In Sec.III, B. The equation (1) is mentioned and well explained in the paper. What does equation (2) do? Need explanation.

**Strengths:**

This paper proposes a novel idea: integrating expert eye movement data into AI for better Glaucoma Diagnosis performance. Several techniques are used to improve AI capability. The paper is well organized and evaluation is very comprehensive.

**Summary Of The Paper:**

The auther aim to improve Glaucoma Diagnosis process with vision transformer together with expert eye movement knowledge. Two major techniques are developed to facilitate AI implementation.

**Weaknesses:**

The paper trained multiple models but with a tiny size of the dataset. And heavily rely on the availability of expert data.

---

### Official Review · Reviewer_yGQx · 2024-08-16
**The paper is of high quality and offers a valuable contribution to the research community, particularly in the application of AI in healthcare and glaucoma diagnosis. I recommend it for acceptance, subject to minor revisions.**

**Overall Rating:** 7
**Confidence:** 4

**Other Quality Metrics:**

(a) Clarity of writing (good)
(b) Clinical Significance (great)
(c) Methodological Novelty (good)
(d) Experiments and Results (great)

**Questions For The Authors:**

Overall, I find your work to be of good quality, with significant potential to aid researchers in glaucoma diagnosis and other related applications. However, the technical details require further clarification, and including a table for the hyperparameters and pseudocode would greatly enhance this aspect.

**Strengths:**

The paper has several strengths:
1. It utilizes a real-time dataset, which often presents challenges such as noisy data, ground truth inconsistencies, and corrupt files, necessitating thorough data cleaning and preprocessing.
2. The proposed models demonstrate strong performance, achieving a mean F1-score of over 92%.
3. The manuscript is well-written and logically structured, facilitating clear communication of the research.

**Summary Of The Paper:**

The article titled "Medical-Expert Eye Movement Augmented Vision Transformers for Glaucoma Diagnosis," in which the authors present two novel approaches to enhance Vision Transformers (ViT) for glaucoma diagnosis by incorporating eye-tracking data. The authors introduced models that leverage eye movement fixation order and attention-alignment loss during training. These Fixation-Order-Informed (FOI) ViT models outperformed the original ViT. Additionally, attention-alignment in the loss function improved performance, particularly by enhancing the impact of clinician-generated spatial attention. Their findings suggest that integrating expert eye-tracking data can significantly boost ViT models' effectiveness in glaucoma diagnosis.

**Weaknesses:**

I have a few suggestions for improvement:
1. Presenting the hyperparameters of the FOI-ViT model in a table would enhance the clarity and ease of interpretation.
2. Including a pseudocode or algorithm for the proposed approach would further improve the paper's presentation and assist readers in understanding the methodology.

---

### Decision · Program_Chairs · 2024-09-23

Accept